# Learning Deeply Shared Filter Bases for Efficient ConvNets

## Abstract

Recently, inspired by repetitive block structure of modern ConvNets, such as ResNets, parameter-sharing among repetitive convolution layers has been proposed to reduce the size of parameters. However, naive sharing of convolution filters poses many challenges such as overfitting and vanishing/exploding gradients, resulting in worse performance than non-shared counterpart models. Furthermore, sharing parameters often increases computational complexity due to additional operations for re-parameterization. In this work, we propose an efficient parameter-sharing structure and an effective training mechanism for recursive ConvNets. In the proposed ConvNet architecture, convolution layers are decomposed into a filter basis, that can be shared recursively, and non-shared layer-specific parts. We conjecture that a shared filter basis combined with a small amount of layer-specific parameters can retain, or further enhance, the representation power of individual layers, if a proper training method is applied. We show both theoretically and empirically that potential vanishing/exploding gradients problems can be mitigated by enforcing orthogonality to the shared filter bases. Experimental results demonstrate that our scheme effectively reduces redundancy by saving up to 63.8% of parameters while consistently outperforming non-shared counterpart networks even when a filter basis is shared by up to 10 repetitive convolution layers.

## 1 Introduction

Modern networks such as ResNets usually have massive identical convolution blocks and recent analytic studies (Jastrzebski et al., 2018) show that these blocks perform similar iterative refinement rather than learning new features. Inspired by these massive identical block structure of modern networks, recursive ConvNets sharing weights across iterative blocks have been studied as a promising direction to parameter-efficient ConvNets (Jastrzebski et al., 2018; Guo et al., 2019; Savarese & Maire, 2019). However, repetitive use of parameters across many convolution layers incurs several challenges that limit the performance of such recursive networks. First of all, deep sharing of parameters might result in *vanishing gradients* and *exploding gradients* problems, which are often found in recurrent neural networks (RNNs) (Pascanu et al., 2013; Jastrzebski et al., 2018). Another challenge is that overall representation power of the networks might be limited by using same filters repeatedly for many convolution layers.

To address aforementioned challenges, in this paper, we propose an effective and efficient parameter-sharing mechanism for modern ConvNets having many repetitive convolution blocks. In our work, convolution filters are decomposed into a fundamental and reusable unit, which is called a *filter basis*, and a layer-specific part, which is called *coefficients*. By sharing a filter basis, not whole convolution filters or a layer, we can impose two desirable properties on the shared parameters: (1) resilience against vanishing/exploding gradients, and (2) representational expressiveness of individual layers sharing parameters. We first show theoretically that a shared filter basis can cause *vanishing gradients* and *exploding gradients* problems, and this problem can be controlled to a large extent by making filter bases orthogonal. To enforce the orthogonality of filter bases, we propose an orthogonality regularization to train ConvNets having deeply shared filter bases. Our experimental results show that the proposed orthogonality regularization reduces the redundancy not just in deeply shared filter bases, but also in none-shared parameters, resulting in better performance than over-parameterized counterpart networks. Next, we make convolution layers with shared parameters more expressive using a hybrid approach to sharing filter bases, in which a small number of layer-specific non-shared

filter basis components are combined with shared filter basis components. With this hybrid scheme, the constructed filters can be positioned in different vector subspaces that reflect the peculiarity of individual convolution layers. We argue that these layer-specific variations contribute to increasing the representation power of the networks when a large portion of parameters is shared.

Since our focus is not on pushing the state-of-the-art performance, we show the validity of our work using widely-used ResNets as a base model on image classification tasks with CIFAR and ImageNet datasets. Our experimental results demonstrate that when each filter basis is shared by up to 10 convolution layers, our method consistently outperforms counterpart ConvNet models while reducing a significant amount of parameters and computational costs. For example, our method can save up to 63.8% of parameters and 33.4% of FLOPs, respectively, while achieving lower test errors than much deeper counterpart models.

Our parameter sharing structure and training mechanism can be applied to modern compact networks, such as MobileNets (Howard et al., 2017) and ShuffleNets (Zhang et al., 2018) with minor adaptations. Since these compact models already have decomposed convolution blocks, some parts of each block can be identified as a shareable filter basis and the rest of the layer-specific parts. In Experiments, we demonstrate that compact MobileNetV2 can achieve further 8-21% parameter savings with our scheme while retaining, or improving, the performance of the original models.

## 2 RELATED WORK

**Recursive networks and parameter sharing:** Recurrent neural networks (RNNs) (Graves et al., 2013) have been well-studied for temporal and sequential data. As a generalization of RNNs, recursive variants of ConvNets are used extensively for visual tasks (Socher et al., 2011; Liang & Hu, 2015; Xingjian et al., 2015; Kim et al., 2016; Zamir et al., 2017). For instance, Eigen et al. (2014) explore recursive convolutional architectures that share filters across multiple convolution layers. They show that recurrence with deeper layers tends to increase performance. However, their recursive architecture shows worse performance than independent convolution layers due to overfitting. In most previous works, filters themselves are shared across layers. In contrast, we propose to share filter bases that are more fundamental and reusable building blocks to construct layer-specific filters.

More recently, Jastrzebski et al. (2018) show that iterative refinement of features in ResNets suggests that deep networks can potentially leverage intensive parameter sharing. Guo et al. (2019) introduce a gate unit to determine whether to jump out of the recursive loop of convolution blocks to save computational resources. These works show that training recursive networks with naively shared blocks leads to bad performance due to the problem of gradient explosion and vanish like RNN (Pascanu et al., 2013; Vorontsov et al., 2017). In order to mitigate the problem of gradient explosion and vanish, they suggest unshared batch normalization strategy. In our work, we propose an orthogonality regularization of shared filter bases to further address this problem.

Savarese & Maire (2019)'s work is also relevant to our work. In their work, the parameters of recurrent layers of ConvNets are generated by a linear combination of 1-2 parameter tensors from a global bank of templates. Though similar to our work, our work suggests more fine-grained filter bases as more desirable building blocks for effective parameter sharing since filter bases can be easily combined with layer-specific non-shared components for better representation power. Our result shows that these layer-specific non-shared components are critical to achieve high performance. Although they achieve about 60% parameter savings, their approach does not outperform counterpart models and incurs slight increases in computational costs due to the overheads in reparameterizing tensors from the templates.

**Model compression and efficient convolution block design:** Reducing storage and inference time of ConvNets has been an important research topic for both resource constrained mobile/embedded systems and energy-hungry data centers. A number of research techniques have been developed such as filter pruning (LeCun et al., 1990; Polyak & Wolf, 2015; Li et al., 2017; He et al., 2017), low-rank factorization (Denton et al., 2014; Jaderberg et al., 2014), quantization (Han et al., 2016), and knowledge distillation (Hinton et al., 2015; Chen et al., 2017), to name a few. These compression techniques have been suggested as post-processing steps that are applied after initial training. Unfortunately, their accuracy is usually bounded by the approximated original models. By contrast, our models are trained from scratch as in Ioannou et al. (2017)'s work and our result shows that

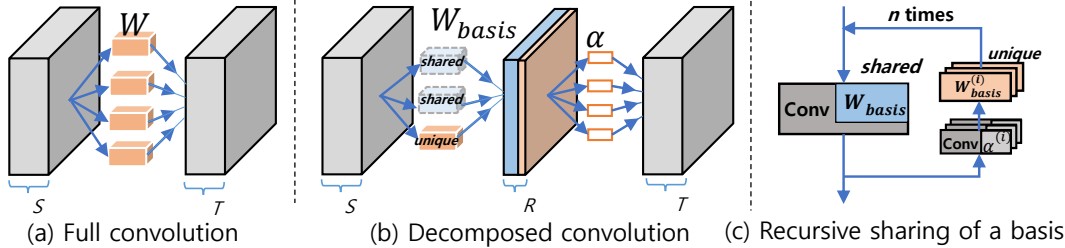

(a) Full convolution  (b) Decomposed convolution  (c) Recursive sharing of a basis

Figure 1: Illustration of the proposed filter basis sharing method. A normal convolution layer in (a) can be decomposed into a filter basis $W_{basis}$ and coefficients $\alpha$ as in (b). A filter basis is a generic building block that can be shared by repetitive convolution layers as in (c). Some layer-specific components ($W_{basis}^{(i)}$ and $\alpha^{(i)}$) can be combined for better expressiveness of recursive layers.

parameter-sharing approaches can outperform the counterpart models, if a proper training method is combined, while achieving significant savings in parameters.

Some compact networks such as ShuffleNet (Zhang et al., 2018) and MobileNet (Howard et al., 2017; Sandler et al., 2018) show that delicately designed internal structure of convolution blocks acquire better ability with lower computational complexity. Our work can be applied to these compact networks since they already exploit decomposed convolutions for their efficient convolution blocks. For instance, since MobileNet's convolution blocks have *pointwise-depthwise-pointwise* convolution steps, the first two convolution steps can be considered a reusable filter basis and the last step can be considered layer-specific coefficients. In Experiments, we show that MobileNetV2 combined with our parameter-sharing scheme outperforms the original models while saving about 8-21% parameters.

## 3  DEEP RECURSIVE SHARING OF A FILTER BASIS

In this section, we discuss how to decompose typical convolution layers into more recursive units, or filter bases, and remaining layer-specific parts. We also discuss how to train ConvNets effectively when filter bases are deeply shared by repetitive convolution layers.

### 3.1  FILTER BASES OF CONVOLUTION LAYERS

We assume that a convolution layer with $S$ input channels, $T$ output channels, and a set of filters $W = \{W_t \in R^{k \times k \times S}, t \in [1..T]\}$. Each filter $W_t$ can be decomposed using a lower rank filter basis $W_{basis}$ and coefficients $\alpha$:

$$W_t = \sum_{r=1}^{R} \alpha_t^r W_{basis}^r, \tag{1}$$

where $W_{basis} = \{W_{basis}^r \in \mathbb{R}^{k \times k \times S}, r \in [1..R]\}$ is a filter basis, and $\alpha = \{\alpha_t^r \in \mathbb{R}, r \in [1..R], t \in [1..T]\}$ is scalar coefficients. In Equation 1, $R$ is the rank of the basis. In a typical convolution layer, output feature maps $V_t \in \mathbb{R}^{w \times h \times T}, t \in [1..T]$ are obtained by the convolution between input feature maps $U \in \mathbb{R}^{w \times h \times S}$ and filters $W_t, t \in [1..T]$. With Equation 1, this convolution can be rewritten as follows:

$$V_t \quad = U * W_t \quad = U * \sum_{r=1}^{R} \alpha_t^r W_{basis}^r \tag{2}$$

$$= \sum_{r=1}^{R} \alpha_t^r (U * W_{basis}^r), \text{ where } t \in [1..T]. \tag{3}$$

In Equation 3, the order of the convolution operation and the linear combination of filter basis is reordered according to the linearity of convolution operators. This result shows that a standard convolution layer can be replaced with two successive convolution layers as shown in Figure 1-(b). The first decomposed convolution layer performs $R$ convolutions between $W_{basis}^r, r \in [1..R]$ and

input feature maps $U$, and it generates an intermediate feature map basis $V_{basis} \in \mathbb{R}^{w \times h \times R}$. The second decomposed convolution layer performs point-wise $1 \times 1$ convolutions that linearly combine $R$ intermediate feature maps $V_{basis}$ to generate output feature maps $V$. The computational complexity of the original convolution is $O(whk^2ST)$ while the decomposed operations take $O(wh(k^2SR + RT))$. As far as $R < T$, the decomposed convolution has lower computational complexity than the original convolution. Due to this computational efficiency, many compact networks such as MobileNets and ShuffleNets also have similar block structures of decomposed convolution layers. For instance, MobileNets have repetitive convolution blocks of *pointwise-depthwise-pointwise* convolutions. The filters in the first two steps can be considered a reusable filter basis and the remaining 1x1 filters in the last step can be considered layer-specific coefficients.

## 3.2 RECURSIVE SHARING OF A FILTER BASIS

In typical ConvNets, convolution layers have different filters $W$s and, hence, each decomposed convolution layer has its own filter basis $W_{basis}$ and coefficients $\alpha$. In contrast, our primary goal in decomposing convolution layers is to share a single filter basis (or a small number of filter bases) across many recursive convolution layers. Unlike some previous works (Jastrzebski et al., 2018; Köpüklü et al., 2019), in which convolution filters $W$ themselves are shared recursively, we argue that a filter basis $W_{basis}$ is a more intrinsic and reusable building block that can be shared effectively since a filter basis constitutes a subspace, in which high dimensional filters across many convolution layers can be approximated.

Though components of a basis only need to be independent and span a vector subspace, some specific bases are more convenient and appropriate for specific purposes. For the purpose of sharing a filter basis, we need to find an optimal filter basis $W_{basis}$ that can expedite the training of filters of shared convolution layers. Although this optimization can be done with a typical stochastic gradient descent (SGD), one problem is that exploding/vanishing gradients problems might prevent efficient search of the optimization space. More formally, we consider a series of $N$ decomposed convolution layers, in which a filter basis $W_{basis}$ is shared $N$ times. Let $\mathbf{x}^i$ be the input of the $i$-th convolution layer, and $a^{i+1}$ be the output of the convolution of $\mathbf{x}^i$ with the filter basis $W_{basis}$

$$a^i(\mathbf{x}^{i-1}) = W_{basis}^\top \mathbf{x}^{i-1}. \tag{4}$$

In Equation 4, $W_{basis} \in \mathbb{R}^{k^2S \times R}$ is a reshaped filter basis that has basis components at its columns. We assume that input $\mathbf{x}$ is properly adapted (e.g., with im2col) to express convolutions using a matrix-matrix multiplication. Since $W_{basis}$ is shared across $N$ recusrive convolution layers, the gradient of $W_{basis}$ for some loss function $L$ is:

$$\frac{\partial L}{\partial W_{basis}} = \sum_{i=1}^{N} \frac{\partial L}{\partial a^N} \prod_{j=i}^{N-1} \left( \frac{\partial a^{j+1}}{\partial a^j} \right) \frac{\partial a^i}{\partial W_{basis}}, \tag{5}$$

, where

$$\frac{\partial a^{j+1}}{\partial a^j} = \frac{\partial a^{j+1}}{\partial \mathbf{x}^j} \frac{\partial \mathbf{x}^j}{\partial a^j} = W_{basis} \frac{\partial \mathbf{x}^j}{\partial a^j} \tag{6}$$

If we plug $W_{basis} \frac{\partial \mathbf{x}^j}{\partial a^j}$ in Equation 6 into Equation 5, we can see that $\prod \frac{\partial a^{j+1}}{\partial a^j}$ is the term that makes gradients unstable since $W_{basis}$ is multiplied many times. This exploding/vanishing gradients can be controlled to a large extent by keeping $W_{basis}$ close to orthogonal (Vorontsov et al., 2017). For instance, if $W_{basis}$ admits eigendecomposition, $[W_{basis}]^N$ can be rewritten as follows:

$$[W_{basis}]^N = [Q\Lambda Q^{-1}]^N = Q\Lambda^N Q^{-1}, \tag{7}$$

where $\Lambda$ is a diagonal matrix with the eigenvalues placed on the diagonal and $Q$ is a matrix composed of the corresponding eigenvectors. If $W_{basis}$ is orthogonal, $[W_{basis}]^N$ neither explodes nor vanishes, since all the eigenvalues of an orthogonal matrix have absolute value 1. Similarly, an orthogonal shared basis ensures that forward signals neither explodes nor vanishes. We also need to ensure that the norm of $\frac{\partial \mathbf{x}^j}{\partial a^j}$ in Equation 5 is bounded (Pascanu et al., 2013) for stability during forward and backward passes. It is shown that batch normalization after non-linear activation at each convolution layer ensures healthy norms (Ioffe & Szegedy, 2015; Guo et al., 2019; Jastrzebski et al., 2018).

For training networks, the orthogonality of shared bases can be enforced with an *orthogonality regularizer*. For instance, when each residual block group of a ResNet shares a filter basis for its convolution layers, the objective function $L_R$ can be defined to have an orthogonality regularizer in addition to the original loss $L$:

$$L_R = L + \lambda \sum_{g}^{G} \|W_{basis}^{(g)}{}^{\top} \cdot W_{basis}^{(g)} - I\|^2, \tag{8}$$

where $W_{basis}^{(g)}$ is a shared filter basis for $g$-th residual block group and $\lambda$ is a hyperparameter.

### 3.3 A HYBRID APPROACH TO SHARING A FILTER BASIS

In our filter basis sharing approach, filters of many convolution layers are constructed by the linear combination of a shared filter basis as in Equation 1. This implies that those high-dimensional filters are all in the same low-dimensional subspace. If the rank of a filter basis is too low, it is very challenging to find such subspace that can express individual peculiarity of many layers' filters. Conversely, if the rank of a shared filter basis is too high (e.g., $R \geq T$), the gain in computational complexity from decomposing filters is mitigated. One way to increase the representational power of each convolution layer, while still maintaining its computational complexity low, is adding a small number of layer-specific components to the filter basis. For instance, we build a filter basis $W_{basis}$ not only using shared components, but also using non-shared components:

$$W_{basis} = W_{bs\_shared} \cup W_{bs\_unique}, \tag{9}$$

where $W_{bs\_shared} = \{W_{bs\_shared}^r \in \mathbb{R}^{k \times k \times S}, r \in [1..n]\}$ are shared filter basis components, and $W_{bs\_unique} = \{W_{bs\_unique}^r \in \mathbb{R}^{k \times k \times S}, r \in [n+1..R]\}$ are per-layer non-shared filter basis components. With this hybrid scheme, filters in different convolution layers are placed in different layer-specific subspace. One disadvantage of this hybrid scheme is that non-shared filter basis components require more parameters. The ratio of non-shared basis components can be varied to control the tradeoffs. But, our results in Section 4 show that only a few per-layer non-shared basis components is enough to achieve high performance.

## 4 EXPERIMENTS

In this section, we perform a series of experiments on image classification tasks. Using ResNets (He et al., 2016) as base models, we train networks with the proposed method and compare them with the baseline networks. We also analyze the effect of the orthogonality regularization and the hybrid scheme.

### 4.1 RESULTS ON CIFAR

#### 4.1.1 MODEL CONFIGURATION AND TRAINING DETAILS

Throughout the experiments, we use ResNets (He et al., 2016) as base networks by replacing their $3 \times 3$ convolution layers to decomposed convolution layers sharing filter bases. Since each residual block group of ResNets have different number of channels and kernel sizes, our networks share a filter basis only in the same group (Figure 5 in Appendix). In each group with $n$ residual blocks, the first block has a different stride, and, hence, it does not share a filter basis. Each residual block of the baseline ResNets has two $3 \times 3$ convolution layers, and, hence, our networks' each group has $2(n-1)$ decomposed convolution layers sharing a filter basis. Throughout the experiments, we denote by ResNet$L$-S$s$U$u$ a ResNet with $L$ layers that has a filter basis with $s$ shared components and $u$ layer-specific non-shared components in the first residual block group. This ratio between $s$ and $u$ is maintained for all residual block groups.

The CIFAR-10/100 datasets contains 50,000 and 10,000 three-channel $32 \times 32$ images for training and testing, respectively. For training networks, we follow a similar training scheme in He et al. (2016). Standardized data-augmentation and normalization are applied to input data. Networks are trained for 300 epochs with SGD optimizer with a weight decay of 5e-4 and a momentum of 0.9. The learning rate is initialized to 0.1 and is decayed by 0.1 at 50% and 75% of the epochs.

### 4.1.2 RESULTS

Table 1 shows the results on CIFAR-100. Networks trained with the proposed method consistently outperform their ResNet counterparts. For instance, ResNet34-S32U1 requires only 36.2% parameters and 66.6% FLOPs of the counterpart ResNet34 while achieving even lower test error (21.84%) than much deeper ResNet50 (22.36%). To show the generality of our work, we apply the proposed method to DenseNet (Huang et al., 2017), ResNeXt (Xie et al., 2017), and MobileNetV2 (Sandler et al., 2018). Although the overall gain is not as great as ResNets', we still observe reduction of resource usages in these networks. For instance, ResNeXt50-S64U4 outperforms the counterpart ResNeXt50 while saving parameters and FLOPs by 16.7% and 12.1%, respectively. In ResNeXt, the gain is limited since they mainly exploit group convolutions; each group convolution is decomposed for filter basis sharing in our network. Similarly, for DenseNet, each $3 \times 3$ convolution layer has a relatively small number of output channels, and, hence the overall gain is not pronounced as much as ResNets'. Unlike the other networks, MobileNetV2 already has a factorized block structure (pointwise-depthwise-pointwise), and, hence, we choose to share the pointwise-depthwise layers across successive convolution blocks with the same number of channels (Figure 7 in Appendix). During the training, the proposed orthogonality regularization is applied to these shared filters separately. Though MobileNetV2 does not have as many repeating blocks as ResNets, we observe that our MobileNetV2-Shared saves about 21.9% of parameters while outperforming the original MobileNetV2.

Table 1: Error (%) on CIFAR-100. '⋆' denotes having 2 shared bases in each group. '†' denotes orthogonality regularization is not applied.

| Baseline | Model | Params | FLOPs | Error |
|---|---|---|---|---|
| ResNet34 | ResNet34 (baseline) | 21.33M | 2.33G | 22.49 |
| | ResNet34-S32U1[†] (ours) | 7.73M | 1.55G | 22.92 |
| | ResNet34-S32U1 (ours) | **7.73M** | **1.55G** | **21.84** |
| DenseNet121 | DenseNet121 (baseline) | 7.05M | 1.81G | 21.95 |
| | DenseNet121-S16U1 (ours) | **5.08M** | **1.43G** | 22.15 |
| ResNeXt50 | ResNeXt50 (baseline) | 23.17M | 2.71G | 20.71 |
| | ResNeXt50-S64U4 (ours) | **19.3M** | **2.38G** | **20.09** |
| MobileNetV2 | MobileNetV2 (baseline) | 2.43M | 0.14G | 27.79 |
| | MobileNetV2-Shared⋆(ours) | **1.90M** | 0.14G | **27.20** |

Table 2: Error (%) on CIFAR-10. '⋆' denotes having 2 shared bases in each group. '†' denotes orthogonality regularization is not applied.

| Baseline | Model | Params | FLOPs | Error |
|---|---|---|---|---|
| ResNet32 | ResNet32 (baseline) | 0.46M | 0.14G | 7.51 |
| | ResNet32-S16U1⋆ (ours) | **0.24M** | 0.16G | **6.95** |
| ResNet56 | ResNet56 (baseline) | 0.85M | 0.25G | 6.97 |
| | ResNet56-S16U1[†] (ours) | 0.27M | 0.30G | 7.70 |
| | ResNet56-S16U1 (ours) | 0.27M | 0.30G | 7.46 |
| | ResNet56-S16U1⋆ (ours) | **0.31M** | 0.30G | **6.33** |
| | Filter Pruning (Li et al., 2017) | 0.77M | 0.18G | 6.94 |
| | KSE (Li et al., 2019) | 0.43M | 0.12G | 6.77 |
| | DR-Res 40 (Guo et al., 2019) | 0.50M | 0.22G | 6.51 |

The result on CIFAR-10 is presented in Table 2. Unlike networks on CIFAR-100, networks on CIFAR-10 has much fewer channels (e.g. 16 channels in the first residual block group) and, hence, projecting filters to such low dimensional subspace might limit the performance of the networks. For instance, in ResNet32-S8U1, filters are supposed to be projected onto 9 dimensional subspace consisting of 8 shared and 1 layer-specific filter basis components. By increasing the rank of filter bases, the better accuracy can be achieved at the cost of increased FLOPs. For deeper networks such as ResNet56, a filter basis is supposed to be shared by many residual blocks in the group, and it can

damage the performance. For example, every filter basis in ResNet56-S16U1 is shared by 8 residual blocks, or 16 convolution layers. Due to this excessive sharing, though ResNet56-S16U1 saves 41.3% parameters, its testing error (7.46%) is higher than the counterpart ResNet56's (6.97%).

To remedy this problem, we introduce a variant, in which each residual block group of the networks uses 2 shared bases; one basis is shared by the first convolution layers of all residual blocks, and the other is shared by the second convolution layers of the same blocks. In Table 2, networks with a '*' mark denote this variant. Though this variant slightly increases the parameters of the networks, it can prevent excessive sharing of parameters. For example, although ResNet56-S16U1* needs 0.04M more parameters for additional shared bases, it still saves 63% parameters of the counterpart ResNet56 and achieves lower testing error, 6.33%.

In Table 2, we compare our results with similar state-of-the-art techniques. Our method achieves better performance and parameter-saving than other approaches such as filter pruning (Li et al., 2017), kernel clustering (Li et al., 2019), and recursive sharing (Guo et al., 2019).

### 4.1.3 ANALYSIS: EFFECTS OF ORTHOGONAL BASES

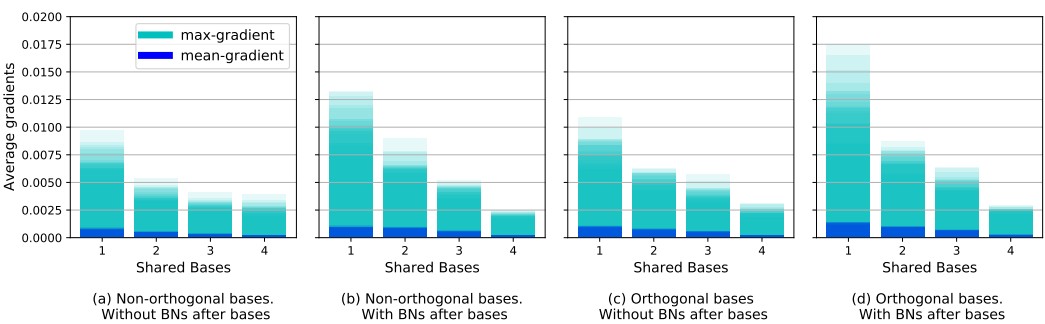

(a) Non-orthogonal bases. Without BNs after bases    (b) Non-orthogonal bases. With BNs after bases    (c) Orthogonal bases Without BNs after bases    (d) Orthogonal bases. With BNs after bases

Figure 2: The flows of gradients in 4 shared bases of ResNet34-S16U1 at the same epoch. For comparison, orthogonal regularization and batch normalization (BN) following the bases are turned on and off. In (b) and (c), BNs and orthogonal regularization, respectively, improve the flow of gradients. In (d), when both BNs and orthogonal regularization are applied simultaneously, the strongest flow of gradients is observed. This trend is consistently observed during the training.

To inverstigate the effect of orthogonality regularization during training, we track the flows of gradients while training ResNet34-S16U1. Figure 2 shows the maximum and mean absolute gradients in the four shared bases during the 20th epoch. Jastrzebski et al. (2018) and Guo et al. (2019) showed that unshared batch normalization (BN) mitigates vanishing/exploding gradients problems, and our result in Figure 2-(b) shows that unshared BNs following shared bases improves the flow of gradients. When the proposed orthogonality regularization is applied to the shared bases, similar effect on gradient flows is observed in Figure 2-(c). When both unshared BN and orthogonality regularization are applied together, in Figure 2-(d), further stronger, but still bounded, flow of gradients are observed. This trend is consistently observed during the training. We conjecture that this healthy flow of gradients improve the optimization process during training.

To further analyze the effect of the orthogonality regularization, in Figure 3, we illustrate absolute cosine similarities of all filter basis components and coefficients of the 2nd and the 3rd residual block groups of ResNet34-S16U1. In the upper low, the X and Y axes display the indexes to the shared basis components first, and all non-shared basis components in respective groups next. As expected, the shared filter basis components have almost zero cosine similarities when the orthogonality regularization in Equation 8 is applied. The bottom low shows the absolute cosine similarities of coefficients of the corresponding groups. In Figure 3, we can clearly see that coefficients manifest lower similarities when the orthogonality regularization is applied. Without the orthogonality regularization, interesting grid patterns are observed in coefficients. This repetitive grid pattern might be related to ResNets' nature of iterative refinement (Jastrzebski et al., 2018). However, since bases and coefficients are used to build filters of recursive layers, such high cosine similarity is directly related to the higher redundancy in the networks. When the orthogonality regularization is applied, such repetitive patterns are less evident, implying that recursive layers perform less repetitive tasks.

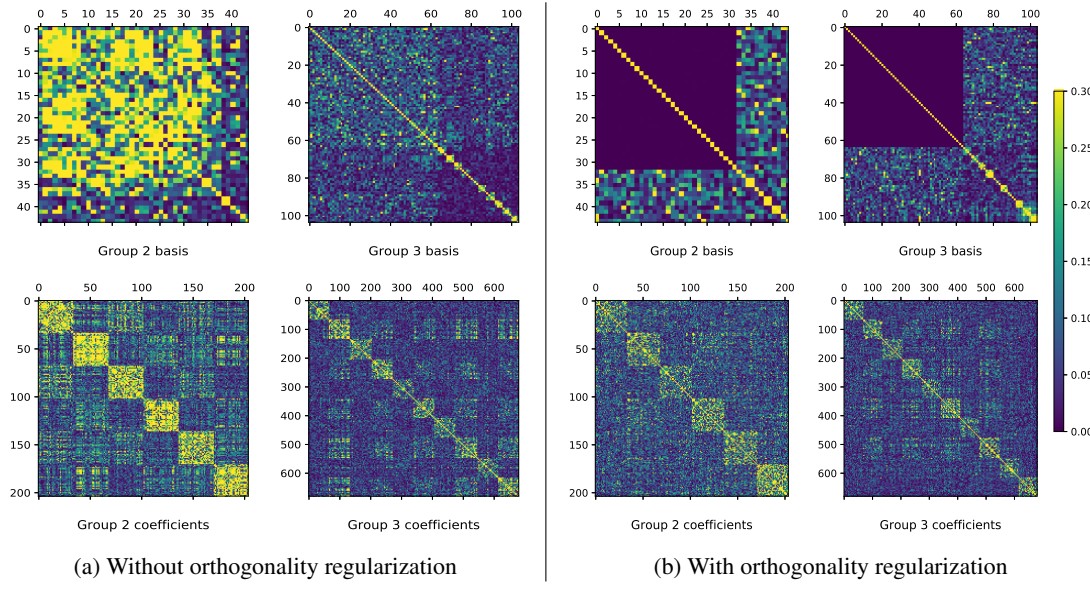

(a) Without orthogonality regularization

(b) With orthogonality regularization

Figure 3: Cosine similarities of bases and coefficients of ResNet34-S16U1 (2-th and 3-th groups.) In the upper row, X and Y axis are indexes to the shared/unique components of the bases. The first 32 and 64 basis components of the 2-th and 3-th groups are shared by 6 and 10 recursive convolution layers, respectively. The others are non-shared unique basis components of those layers. Orthogonality regularization is applied only to shared components. The lower row shows corresponding coefficients in the residual block groups. Brighter color corresponds to higher similarity.

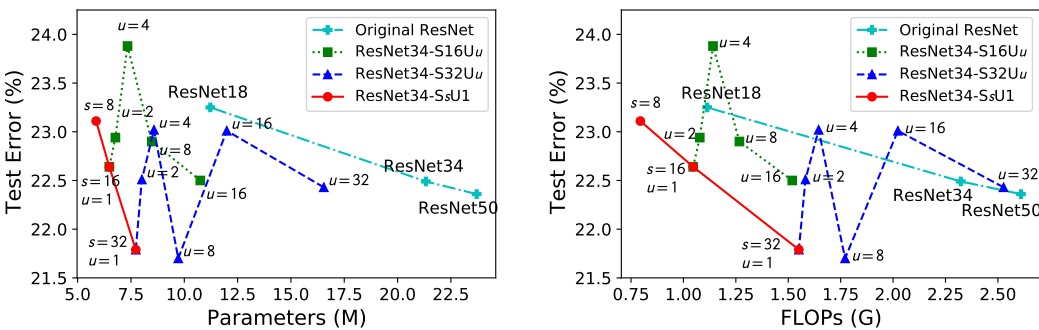

Figure 4: Testing errors vs. the number of parameters and FLOPs on CIFAR-100. The number of shared basis components ($s$), and non-shared basis components ($u$) are varied. Using more shared basis components results in better performance. In contrast, using more non-shared components does not always improve performance.

### 4.1.4 ANALYSIS: EFFECTS OF RANKS OF SHARED/UNSHARED BASES

Figure 4 shows test errors as parameters and FLOPs are increased by varying the number of shared/non-shared basis components of networks. In general, the higher performance is expected with the more parameters. We observe that this presumption is true for shared basis components. For instance, when the number of shared basis components $s$ is varied from 8 to 32, the test error sharply decreases from 23.1% to 21.7%. However, non-shared basis components manifest counter-intuitive results. Although a small number of non-shared basis components (e.g., $u = 1$) are clearly beneficial to the performance, the higher $u$'s do not always lead to the higher performance. For instance, when $u = 4$, both ResNet34-S16U$u$ and ResNet34-S32U$u$ show the worst performance. This result demonstrates the difficulty of training networks with larger parameters. Further study is required for this problem.

## 4.2 RESULTS ON IMAGENET

We evaluate our method on the ILSVRC2012 dataset (Russakovsky et al., 2015) that has 1000 classes. The dataset consists of 1.28M training and 50K validation images. We use ResNet34/50 and MobileNetV2 as base models. Since the block structure of ResNet50 and MobileNetV2 already resembles the decomposed convolution blocks, our models derived from ResNet50 and MobileNetV2 share the first 1x1 and 3x3 convolution layers recursively without filter decomposition for repeating convolution blocks (Figure 6 and 7 in Appendix). We train the ResNet-derived models for 150 epochs with SGD optimizer with a mini-batch size of 512, a weight decay of 1e-4, and a momentum of 0.9. The learning rate starts with 0.1 and decays by 0.1 at 60-th, 100-th, and 140-th epochs. MobileNetV2 and our MobileNetV2-Shared models are trained for 300 epochs with a weight decay of 1e-5. Its learning rate starts with 0.1 and decays by 0.1 at 150-th, 225-th, and 285-th epochs.

Table 3: Error (%) on ImageNet. '$\star$' denotes having 2 shared bases in each residual block group. In MobileNetV2-Shared[†], the first 1x1 step of each block shares parameters recursively. Latency is measured on Nvidia Jetson TX2 (GPU, batch size = 1).

| Baseline | Model | Params | FLOPs | top-1 | top-5 | Latency |
|---|---|---|---|---|---|---|
| ResNet34 | ResNet34 (baseline) | 21.80M | 7.34G | 26.70 | 8.58 | 33.6ms |
| | ResNet34-S32U1 (ours) | 8.20M | 4.98G | 27.83 | 9.42 | 31.6ms |
| | ResNet34-S48U1$^\star$ (ours) | **11.79M** | **6.52G** | **26.67** | **8.54** | 38.6ms |
| | Filter Pruning (Li et al., 2017) | 19.30M | 5.52G | 27.83 | - | - |
| ResNet50 | ResNet50 (baseline) | 25.56M | 8.22G | 23.85 | 7.13 | 43.8ms |
| | ResNet50-Shared (ours) | 18.26M | 8.22G | 23.95 | 7.14 | 43.5ms |
| | FSNet (Yang et al., 2020) | 13.9M | - | 26.89 | 8.63 | |
| MobileNetV2 | MobileNetV2 (baseline) | 3.50M | 0.66G | 28.0 | 9.71 | 18.4ms |
| | MobileNetV2-Shared[†] (ours) | 3.24M | 0.66G | **27.61** | **9.34** | 17.9ms |
| | MobileNetV2-Shared (ours) | **2.98M** | 0.66G | 28.21 | 9.85 | 17.8ms |
| | FBNet-A (Wu et al., 2019) | 4.3M | 0.49G | 27.0 | - | - |
| | DR-MobileNetV2 (Guo et al., 2019) | 2.96M | 0.53G | 28.2 | 9.72 | - |

The results in Table 3 show that ResNet34-S48U1$^\star$ outperforms the counterpart ResNet34 while using only 54.0% parameters of the counterpart. Although ResNet34-S48U1$^\star$ requires lower FLOPs than the counterpart ResNet34, it takes 14% longer latency on Jetson TX2. This overhead mostly comes from performing convolution operations separately for shared and non-shared basis components.

Since our ResNet50- and MobileNetV2-derived models do not decompose convolution blocks to define shared bases, overall parameter-saving is not as pronounced as in ResNet34. However, they still save about 28.6% and 14.9% parameters, respectively, while achieving comparable performance. For example, MobileNetV2-Shared[†] even outperforms the counterpart model by 0.51%. This result shows that even compact models such as MobileNetV2 can benefit from the proposed parameter-sharing mechanism using orthogonality regularization. Since our ResNet50- and MobileNetV2-derived models do not perform separate convolution for shared and non-shared basis, their theoretical FLOPs are equal to the counterpart models'. However, the latency of our parameter-efficient models is slightly lower than the counterpart models on actual devices. This is because our parameter-efficient models better make use of the limited cache and memory of the embedded device.

## 5 CONCLUSIONS AND FUTURE WORKS

In this work, we propose to share filter bases of decomposed convolution layers for efficient and effective sharing of parameters in ConvNets. We both theoretically and empirically show that gradient explosion/vanishing problem of deeply shared filter bases can be effectively addressed by the proposed orthogonality regularization of the filter bases. With our approach, while a significant amount of parameters is shared, the representation power of each convolution layer is retained, or further enhanced, by reducing the redundancy effectively. Experimental results show that our approach consistently outperforms the counterpart models while significantly reducing parameters. We believe that the proposed parameter-sharing and training method suggests important architectural possibilities for future neural architecture search (NAS) for resource-efficient networks.

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

# A APPENDIX

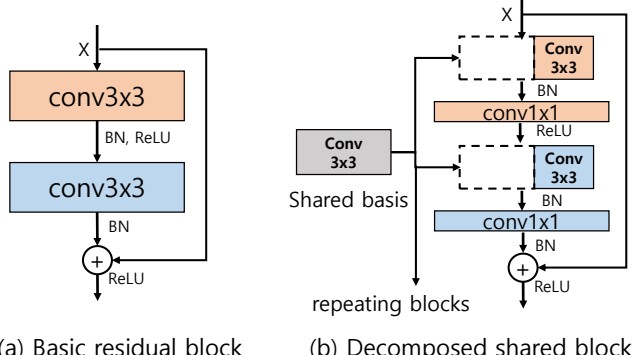

(a) Basic residual block      (b) Decomposed shared block

Figure 5: Block structure of ResNet34 and a shared basis. For basic blocks, a shared basis is defined by decomposing original 3x3 convolution filters.

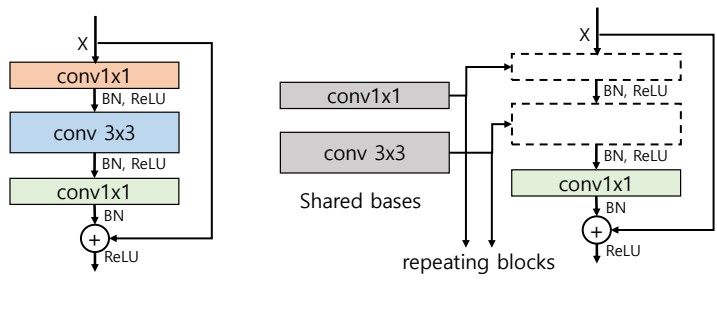

(a) Bottleneck residual block      (b) Shared bottleneck block

Figure 6: Block structure of ResNet50 and shared bases. In the shared bottleneck block, the first 1x1 convolution and the second 3x3 convolution are considered as two bases of the bottleneck block. During training, orthogonal regularization is applied to these bases separately.

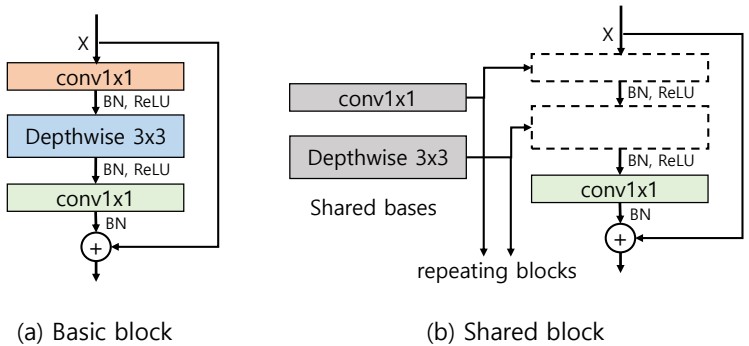

(a) Basic block      (b) Shared block

Figure 7: Block structure of MobileNetV2 and shared bases. In the shared block, the first 1x1 pointwise convolution and the second 3x3 depthwise convolution are considered as two bases of the block. During training, orthogonal regularization is applied to these bases separately.

