# OpenReview forum: "Learning Deeply Shared Filter Bases for Efficient ConvNets"
_ICLR.cc/2021/Conference — Reject_

### Official Review · AnonReviewer4 · 2020-10-28
**The proposed method achives outstanding performance on ResNet-34, while I have concerns on main contribution and experiments**

**Rating:** 5
**Confidence:** 3

**Review:**

To compress a DNN model, the proposed method improves traditional kernel sharing methods by sharing kernel bases obtained from decomposing a convolution layer. Experiments on ImageNet show that the method reduces 10.01M parameters of ResNet-34 with almost the same error rate, i.e., the method is more effective for DNN architectures without using separable convolution.

pros.
1. In Equations (1)~(3), the authors clearly and formally explain one of their motivations that a convolution layer with high computing complexity can be replaced by a convolution layer with lower computing complexity and a liner layer.
2. A classic problem, exploding/vanishing gradients, is alleviated by keeping a variable W_basis close to orthogonal. Moreover, in Section 4.1.3, experiments without orthogonality and with orthogonality are qualitatively and quantitatively compared.
3. In Table 3, the proposed method reduces 10.01M parameters of ResNet-34 with almost the same error rate.

cons.
1. The novelty/contribution is not quite sufficient. The main novely of this paper is to simultaneously train convolution decomposing and weight sharing. The strategy of convolution decomposing (as shown in Equation 3) in this paper is similar with separable convolution used in MobileNet. For weight sharing, the authors propose to share weight across layers which is also similar with previous weight sharing papers.
2. The experiments are not sufficient. In Section Experiments, the proposed method is only compared with classic ResNet-34 and MobileNet-V2. However, tons of traditional DNN compressiong papers work on decomposing a convolutional layer, matrix decomposing, or sharing weights. The proposed method should be compared with these traditional methods.

---

> ### Author Response · Authors · 2020-11-21
> **Thank you for your review!**
>
> **Q:**
> 1.The novelty/contribution is not quite sufficient. The main novelty of this paper is to simultaneously train convolution decomposing and weight sharing. The strategy of convolution decomposing (as shown in Equation 3) in this paper is similar with separable convolution used in MobileNet. For weight sharing, the authors propose to share weight across layers which is also similar with previous weight sharing papers.
>
> **Answer:**
> We acknowledge that neither filter decomposition nor recursive parameter sharing have novelty. However, any one of them or combination of them have achieved comparable performance to ours. Unlike previous filter decomposition works, we focus only on the block structure of the decomposed filters to extract sharable bases and we further claim that if we enforce some desirable properties, such as orthogonality, to these shared bases, we can reduce the redundancy of original networks while achieving better performance. Hence our major contribution is not in filter decomposition, but in novel training approach to recursively shared bases. In the original submission, we showed only theoretically that orthogonal bases mitigate vanishing /exploding gradients and did not support our claim with experiments. To further support our contribution experimentally, we add Section 4.1.3 and Figure 2 to show that orthogonal bases improve the flow of gradients. Figure 2 shows that the proposed orthogonality regularization has similar effect as batch normalization on gradient flows. This healthy flow of gradients improves the optimization process during training and enables our models achieve superior performance with less parameters.
> ---
>
> **Q:**
> 2.The experiments are not sufficient. In Section Experiments, the proposed method is only compared with classic ResNet-34 and MobileNet-V2. However, tons of traditional DNN compressiong papers work on decomposing a convolutional layer, matrix decomposing, or sharing weights. The proposed method should be compared with these traditional methods.
>
> **Answer:**
> Although our approach achieves state-of-the-art performance, in the original submission, we did not compare our result properly with previous works due to page limits. We update results in Tables 1-3 as follows. First, for CIFAR10 & ImageNet experiments, we compare our work with state-of-the-art previous works. Though our approach is simple, the results clearly show that our approach excels those SOTA approaches in most cases. Second, for CIFAR100 & ImageNet experiments, we add new results on MobileNetV2 and ResNet50 to show that shared orthogonal bases alone without factorization still saves much parameters while maintaining, or further increasing, the performance. For example, our ResNet50 saves 28.6% parameters only through recursive sharing of parameters while losing only 0.1% top-1 (and 0.01% top5) accuracy. Due to limited time and resources, we obtained this result without any hyperparameter tuning, and, hence, better performance could be achieved with further tuning. Finally, we improved the visibility of Tables 1-3 to better show our results.
> ---

---

### Official Review · AnonReviewer3 · 2020-10-28

**Rating:** 5
**Confidence:** 4

**Review:**

The authors proposed a parameter-sharing method among repetitive convolution layers, where typical filters are decomposed into a set of resuable filter bases and coefficients. The experimental results show some improvement about the number of parameters and FLOPs. Generally, the paper is well written and the method is presented clearly. The authors also claimed that orthogonality regularization can reduce the potential vanishing/exploding gradients problem in weight sharing training.
Here are my concerns:
(1) The experiments mainly compared the proposed method with non-shared net designs. However, the comparison results or methods/algorithms with other weight sharing paper are lacking. (e.g., Jastrzebski et al.,2018; Köpüklü et al., 2019 mentioned in the paper). Therefore, it's not easy to judge the novelty of this incremental work.
(2) Some of the non-shared baseline accuracy is not as good as reported in previous works (e.g., CIFAR100 accuracy-parameter, MobileNetV2 accuracy.). Some comparative results are not fair enough, for example, in the ImageNet experiments, MobileNetV2-Shared model is trained for 300 epochs but the baseline is only trained for 140 epochs.
(3) The author should, I suggest, focus more on the actual performance such as memory and time overheads during network training and deployment inference, rather than theoretical MACs and FLOPs, (also discussed in ShuffleNet_v2 paper ) which can contribute more to the compact NN community.

---

> ### Author Response · Authors · 2020-11-21
> **Thank you for your review!**
>
> **Q:**
> (1) The experiments mainly compared the proposed method with non-shared net designs. However, the comparison results or methods/algorithms with other weight sharing paper are lacking. (e.g., Jastrzebski et al.,2018; Köpüklü et al., 2019 mentioned in the paper). Therefore, it's not easy to judge the novelty of this incremental work.
>
> **Answer:**
> Although our approach achieves state-of-the-art performance, in the original submission, we did not compare our result properly with previous works due to page limits. We update results in Tables 1-3 as follows. First, for CIFAR10 & ImageNet experiments, we compare our work with state-of-the-art previous works. Although our approach is simple, the results clearly show that our approach excels those SOTA approaches in most cases. Second, for CIFAR100 & ImageNet experiments, we add new results on MobileNetV2 and ResNet50 to show that shared orthogonal bases alone without filter factorization still saves much parameters while maintaining, or further increasing, the performance. For example, our ResNet50 saves 28.6% parameters only through recursive sharing of parameters while losing only 0.1% top-1 (and 0.01% top5) accuracy. Due to limited time and resources, we obtained this result without any hyperparameter tuning, and, hence, better performance could be achieved with further tuning. Finally, we improved the visibility of Table 1~3 to better show our results.
> ---
>
> **Q:**
> (2) Some of the non-shared baseline accuracy is not as good as reported in previous works (e.g., CIFAR100 accuracy-parameter, MobileNetV2 accuracy.).
>
> **Answer:**
> We trained both baselines and our models in the same settings. For example, both the vanilla MobileNetV2 and our model were trained for 300 epochs in the same settings. We know that some efficient networks such as EPSNetV2(CVPR'19) use more sophisticated learning rate scheduler (e.g., cosine learning rate scheduler) to get better results. However, we do not choose to use such recent techniques to exclude the effects coming from different training settings.
> ---
>
> **Q:**
> Some comparative results are not fair enough, for example, in the ImageNet experiments, MobileNetV2-Shared model is trained for 300 epochs but the baseline is only trained for 140 epochs.
>
> **Answer:**
> We trained both baselines and our models in the same settings. We acknowledge that, in the original submission, it was not clearly stated for ImageNet experiments. In the revision, we explicitly mention that "MobileNetV2 and our MobileNetV2-Shared models are trained for 300 epochs with a weight decay of 1e-5."
> ---
>
> **Q:**
> (3) The author should, I suggest, focus more on the actual performance such as memory and time overheads during network training and deployment inference, rather than theoretical MACs and FLOPs, (also discussed in ShuffleNet_v2 paper) which can contribute more to the compact NN community.
>
> **Answer:**
> For ImageNet results (Section 4.2), we provide wall-clock speed measured on Nvidia Jetson TX2. The result is mixed. For ResNet34-derived models (ResNet34-S48U1*), although our model has lower FLOPs than the counterpart ResNet34, it takes 14% longer latency on Jetson TX2. This overhead mostly comes from performing convolution operations separately for shared and non-shared basis components. This requires further optimization to achieve actual speedup. For ResNet50- and MobileNetV2-derived models, our parameter-efficient models show slightly lower latency (~3.3%). We conjecture that this is because our parameter-efficient models better make use of the limited cache and memory of the embedded device.
> ---

---

### Official Review · AnonReviewer2 · 2020-10-28
**Nice solution to decrease computation complexity in CNNs. Needs more work on results section (writing + experiments)**

**Rating:** 6
**Confidence:** 3

**Review:**

[Summary]
This method proposes to decompose convolutional filters using a low rank filter basis where the convolutional operation in a layer consists of shareable filter basis and non-shareable layer coefficients. This is developped to save computational costs whilst maintaining performance. To regularise against vanishing/exploding gradients and promoting more useful representations, the authors seek orthogonal filter basis'. This filter basis is shared across layers in contrast to other works who look at recursive sharing.

[Main Comments]
1. I like the idea to decompose convolutions into shared weights and layer-specific components. This builds in nicely with a lot of work in multi-task learning about learning which weights to share and so I wonder about how your method might generalise to other problems - could the authors comment on this?
2. The authors compared their method against ResNet, MobileNet and DenseNet. I would like to see comparisons against more similar techniques such as [Savarese & Maire 2019] and [Guo et al. 2019] to better appreciate the performance of the method. The results against ResNet/MobileNet are indeed promising and good to see.
3. The presentation of the results (Table 1 and Table 2) is cumbersome. I would suggest reworking this section to make it easier to appreciate the results.
4. The results in general seem good but due to presentation issues, it is difficult to discern what is going on. In particular, the nomenclature for ResNetL-SsUu is confusing.

[Other]
1. Could you add an extra parameter to the filter basis, to control dilation of the convolution and thus learn when it is needed to increase the receptive field in addition to sharing weights?
2. There is a typo in in the sentence above Equation (5) ; reusrive --> recursive
3. I would recommend a figure to display the differences between vanilla ResNet and your version
4. Was it possible to analyse empirically the gradient magnitude to show that the orthogonality regularisation helped?
5. For Figure 2 - I would potentially make a heatmap style figure with $u$ and $s$ as the x-y axis to help display results
6. How does orthogonality regularisation work on a 1x1 conv?
7. The work of Ioannou et al. (https://arxiv.org/abs/1605.06489) might be of interest as this seeks structured sparsity in ConvNets to decrease computation without loss of performance

---

> ### Author Response · Authors · 2020-11-21
> **Thank you for your review!**
>
> **Q:**
> I like the idea to decompose convolutions into shared weights and layer-specific components. This builds in nicely with a lot of work in multi-task learning about learning which weights to share and so I wonder about how your method might generalize to other problems - could the authors comment on this?
>
> **Answer:**
> Unlike our work, multi-task learning works at an inter-network scale. However, in a broader sense, our work has similarity to multi-task learning since multi-task learning also has some shared parts across tasks and some task-specific parts. Although they are working on different scales, we believe, they can inspire each other.  In our work, one major problem is finding desirable properties of shared bases and we show that orthogonality of bases improves the flow of gradients and makes layer-specific parts more distinctive. Similarly, for multi-task learning, we first need to think about what would be (quantifiable) desired properties to learn compact shared representation across tasks.
> ---
> **Q:**
> The authors compared their method against ResNet, MobileNet and DenseNet. I would like to see comparisons against more similar techniques such as [Savarese & Maire 2019] and [Guo et al. 2019] to better appreciate the performance of the method. The results against ResNet/MobileNet are indeed promising and good to see.
>
> **Answer:**
> Although our approach achieves state-of-the-art performance, it was hard to appreciate the results in the original submission. Hence, we update Tables 1-3 to include similar state-of-the-art techniques including [Guo et al. 2019]. Since [Savarese & Maire 2019] uses Wide ResNets as their baseline, we cannot compare their work directly. Hence, we summarize their result in the Related Work section. As mentioned in the Related Work, unlike our work, their work does not outperform baseline models and incurs slight increase of FLOPs.
> Further, for CIFAR100 & ImageNet, we add new results on MobileNet and ResNet50 to show that shared orthogonal bases alone without filter factorization still can save much parameters while maintaining, or further increasing, the performance.
> We also improved the visibility of Tables 1~3 by grouping related techniques according to their baseline models.
> ---
> **Q:**
> . Could you add an extra parameter to the filter basis, to control dilation of the convolution and thus learn when it is needed to increase the receptive field in addition to sharing weights?
>
> **Answer:**
> In our initial design, we tried to make a versatile basis from a smaller set of shared parameters (or templates) that can be used across all layers. However, we could not achieve competitive results. Based on our experience, providing too much versatility or generality hurts performance. Although we have not tested our work for dilated convolution, we'd better define separate filter bases for different dilated convolutions.
> ---
>
> **Q:** I would recommend a figure to display the differences between vanilla ResNet and your version
>
> **Answer:**
>  We add Figures 5~7 in Appendix to show the differences between vanilla models and our models. Due to space limitation, we put them in Appendix.
> ---
>
> **Q:** Was it possible to analyse empirically the gradient magnitude to show that the orthogonality regularisation helped?
>
> **Answer:**
> In the original submission, we showed only theoretically that orthogonal basis mitigates vanishing /exploding gradients and did not support our claim with experiments. To support our claim experimentally, we add Section 4.1.3 and Figure 2 to show that orthogonal bases improve the flow of gradients. Jastrzebski et al. (2018) and Guo et al. (2019) already showed that unshared batch normalization (BN) mitigates vanishing/exploding gradients problems, and our result in Figure 2 shows that the proposed orthogonality regularization has similar effect on gradient flows as batch normalization. This healthy flow of gradients improves the optimization process during training and enables our models achieve superior performance with less parameters.
> ---
> **Q:** How does orthogonality regularisation work on a 1x1 conv?
>
> **Answer:**
> For ResNet50 and MobileNetV2, we already use orthogonality regularization on 1x1 convolutions. Please see Figures 6 and 7 in Appendix for detailed block structure. As shown in results (Table 3), orthogonal bases without factorization (e.g., 1x1 conv.) still saves much parameters while maintaining the performance.
> ---
>
> **Q:** The work of Ioannou et al. (https://arxiv.org/abs/1605.06489) might be of interest as this seeks structured sparsity in ConvNets to decrease computation without loss of performance
>
> **Answer:**
>  Ioannou et al.'s work inspired many later works including ours. As Ioannou et al.'s work, we are not approximating an existing model's weights but creating new network architecture with shared bases, which is then trained from scratch. We add above discussion in Related Work.

---

### Official Review · AnonReviewer1 · 2020-11-02
**Ok but not good enough**

**Rating:** 4
**Confidence:** 4

**Review:**

This paper addresses the problem of obtaining more compact CNNs by a parameter sharing method. The authors propose to represent a weight filter in a low-rank subspace (represented as a linear combination of low-rank filter basis) plus a set of non-shared low-rank filter basis (per-layer). In this way, the shared low-rank filter basis is reused across several layers, and the non-shared ones per layer are used to enhance model generalization ability. Experiments are performed on CIFAR and ImageNet datasets, using some popular CNN structures for evaluation.

This paper suffers from following issues.

---Novelty issue.

To the best of my knowledge, representing a weight filter (in a specific convolutional layer) as a linear combination of a low-rank filter basis was explored in many works, such as “Efficient and Accurate Approximations of Nonlinear Convolutional Networks”, in CVPR 2015, and “Speeding up Convolutional Neural Networks with Low Rank Expansions”, in BMVC 2014. Additionally, even  the first two parts of Figure 1 are very similar to Figure 1 of CVPR 2015 paper. However, they are either missed or not discussed/compared.

Compared with existing low-rank approximation related works, the main contribution of this paper is sharing low-rank filter basis across several layers (in the same group or block of a CNN), while retaining a non-shared low-rank filter basis per layer. To me, the motivation is not clear enough, e.g., such kind of recursive design may easily lead to more high computational cost; it is usually not useful to get obvious benefits. Furthermore, from the definition in the method part, sharing low-rank filter basis across layers needs weight filters have the same shape size. Such a strong constraint also limits its applications.  Additionally, how to determine shared/non-shared low-rank filter basis for different CNNs and their building blocks is not clear enough, and they are manually tuned one by one.

The authors claim that parameter sharing in this way can address “… a shared filter basis can cause vanishing gradients and exploding gradients problems”.  However, there is no convincing experiments to support this claim, e.g., how this happens in the experiments and how this is alleviated?

--- Experiments issue.

Comparison with related works such as existing low-rank methods and recursive parameter sharing methods are completely missing.

Real wall-clock speed comparison is also missing.

On small dataset like CIFAR, experiments are run only once. Due to random effects, it may easily lead to different conclusions. For fair comparison, the authors should at least run experiments for several runs (e.g., 3/5) and report mean accuracies.

Also, on CIFAR-100 dataset, the authors use ResNet34/50/MobileNetV2. However, these CNNs are defined on ImageNet (larger image size 224x224), which have different numbers of down-sampling layers compared to those (ResNet32/56/110) for CIFAR dataset (small image size 32x32). What kind of modifications did the authors make?  Why not using more standard CNNs like  those used on CIFAR-10 to CIFAR-100?

---

> ### Author Response · Authors · 2020-11-21
> **Thank you for your review! (Part II)**
>
> **Q:** Comparison with related works such as existing low-rank methods and recursive parameter sharing methods are completely missing.
>
> **Answer:**
> Although our approach achieves state-of-the-art performance, in the original submission, we did not compare our result properly with previous works due to page limits. We update results in Table 1~3 to compare our work with state-of-the-art works, including pruning, kernel clustering and recursive parameter sharing. Though our approach is simple, the results clearly show that our approach excels those SOTA approaches in most cases.
> ---
> **Q:** Real wall-clock speed comparison is also missing.
>
> **Answer:**
> For ImageNet results (Section 4.2), we provide wall-clock speed measured on Nvidia Jetson TX2. The results are mixed. For ResNet34-derived models (ResNet34-S48U1*), although our model has lower FLOPs than the counterpart ResNet34, it takes 14% longer latency. This overhead mostly comes from performing convolution operations separately for shared and non-shared basis components. This requires further optimization to achieve better speedup.
> For ResNet50- and MobileNetV2-derived models, our parameter-efficient models show slightly lower latency (~3.3%). This is because our parameter-efficient models better make use of the limited cache and memory of the embedded device.
> ---
>
> **Q:**
> On small dataset like CIFAR, experiments are run only once. Due to random effects, it may easily lead to different conclusions. For fair comparison, the authors should at least run experiments for several runs (e.g., 3/5) and report mean accuracies.
>
> **Answer:**
> In the original submission, many results on CIFAR10/100 were already from at least 3 runs. In the revision, the results on CIFAR10/100 are updated to have mean accuracy for all results. Although the updated mean accuracies of some results are lower than before, our result still outperforms the counterpart baselines and competing approaches.
> ---
>
> **Q:**
> Also, on CIFAR-100 dataset, the authors use ResNet34/50/MobileNetV2. However, these CNNs are defined on ImageNet (larger image size 224x224), which have different numbers of down-sampling layers compared to those (ResNet32/56/110) for CIFAR dataset (small image size 32x32). What kind of modifications did the authors make? Why not using more standard CNNs like those used on CIFAR-10 to CIFAR-100?
>
> **Answer:**
> For ResNet, official CNN designs for CIFAR dataset (e.g., ResNet32/56/110) are available in the original paper (Kaming He, 2015), but other models such as MobileNetV2 do not have official design for CIFAR dataset. Hence, we choose to use CNN models designed for ImageNet dataset by adapting strides and kernel sizes. For example, for MobileNetV2, we change the kernel size and stride of the first convolution layer from 3x3 and 3 to 1x1 and 1, respectively. We also change the stride of the last convolution layer from 2 to 1. We referenced several representative Pytorch implementation of the models (e.g., https://github.com/weiaicunzai/pytorch-cifar100)
> Another reason that we use larger (image size 224x224) models for CIFAR100 is that models for CIFAR10 (e.g., ResNet32/56/100) have small number of channels (e.g., 16 in the first residual block group). Hence, factorizing such low dimensional filters to a low-rank basis is not easy. To obtain satisfactory performance, we have to increase the rank of the basis.

---

> ### Author Response · Authors · 2020-11-21
> **Thank you for your review! (Part I)**
>
> **Q:**
> Novelty issue
>
> **Answer:**
> Decomposition of filters is a well-known technique and many papers, such as mentioned CVPR2015 and BMVC2014, exploits filter decomposition in order to approximate existing filters. Our work do not claim filter decomposition as our contribution. Unlike these works, we focus only on the block structure of the decomposed filters to expose a sharable basis and we claim that if we enforce some desirable properties, such as orthogonality, to these shared bases, we can reduce the redundancy of original networks while achieving better performance. If convolution blocks already have a form of decomposed filters, as in MobileNetV2, we can apply our training method directly without filter decomposition. For example, in MobileNetV2 and ResNet50, we consider the first 1x1 and 3x3 convolutions of each block as sharable bases and make them orthogonal during training. (Please see Figure 5~7 in Appendix for detailed block designs.) Though both low-rank approximation of convolution filters and recursive sharing of parameters have been explored in many previous works, any one of them or combination of them have achieved comparable performance to ours.
> ---
>
> **Q:** Furthermore, from the definition in the method part, sharing low-rank filter basis across layers needs weight filters have the same shape size. Such a strong constraint also limits its applications.
>
> **Answer:**
> We acknowledge that this limitation is inevitable in recursive sharing of parameters since recursive layers are supposed to have the same structure. However, we believe that since many modern ConvNets have many repetitive layers with the same shape of weights, there are still lots of chance to save parameters by sharing parameters among them.
> ---
> **Q:** Additionally, how to determine shared/non-shared low-rank filter basis for different CNNs and their building blocks is not clear enough, and they are manually tuned one by one.
>
> **Answer:**
> Providing non-shared components to a filter basis is one way to differentiate recursive layers. We acknowledge that it needs some manual tuning to determine how many such non-shared components are enough to achieve sufficient performance. For example, our results with MobileNetV2 show that no non-shared components of a filter basis is required to achieve comparable performance to the original MobileNetV2. In future work , we might consider including the level of non-shared components in the search space for neural architecture search (NAS).
> ---
> **Q:**: The authors claim that parameter sharing in this way can address “… a shared filter basis can cause vanishing gradients and exploding gradients problems”. However, there is no convincing experiments to support this claim, e.g., how this happens in the experiments and how this is alleviated?
>
> **Answer:**
> Our major contribution is in novel approach to training shared bases with orthogonality regularization. In the original submission, we showed only theoretically that an orthogonal basis mitigates vanishing /exploding gradients and did not support our claim in experiments. To support our contribution experimentally, we add Section 4.1.3 and Figure 2 to show that orthogonal bases improve the flow of gradients. Jastrzebski et al. (2018) and Guo et al. (2019) already showed that unshared batch normalization (BN) mitigates vanishing/exploding gradients problems, and our result in Figure 2 shows that the proposed orthogonality regularization has similar effect on gradient flows. This healthy flow of gradients improves the optimization process during training and enables our models achieve superior performance with less parameters.
> ---

---

### Author Response · Authors · 2020-11-21
**Summary of Revision**

We authors sincerely thank the reviewers for their constructive and valuable comments. We carefully revised our paper according to reviewers' comments. For the response to each reviewer's comments, please see below. In summary, we focus on addressing 2 major concerns raised by the reviewers.

**1. Novelty:**
Some reviewers mention that our work lacks novelty since both low-rank approximation of convolution filters and recursive sharing of parameters have been explored in many previous works. We acknowledge that it is true. However, any one of them or combination of them have achieved comparable performance to ours. Unlike previous low-rank approximation works, we focus only on the block structure of the decomposed filters to extract sharable bases. We claim that if we enforce some desirable properties, such as orthogonality, to these shared bases, we can reduce the redundancy of original networks while achieving better performance. Hence our major contribution is not in filter decomposition, but in novel training approach to shared bases. In the original submission, we showed only theoretically that orthogonal bases mitigate vanishing /exploding gradients. To support our claim further with experiments, we add Section 4.1.3 and Figure 2 to show that orthogonal bases improve the flow of gradients. Figure 2 shows that the proposed orthogonality regularization has similar effects on the flow of gradient as batch normalization. We conjecture that this healthy flow of gradients improves the optimization process during training and enables our models achieve superior performance with less parameters.

**2. Experiments:**
Although our approach achieves state-of-the-art performance, in the original submission, we did not compare our result with previous works properly due to page limits. In the revision, we update results in Table 1~3 as follows.
1) For CIFAR10 & ImageNet, we compare our work with state-of-the-art previous works. Although our approach is simple, the results clearly show that our approach excels those SOTA approaches in most cases. For example, our ResNet56-variant model consumes at least 28% less parameters than competing techniques (pruning, kernel clustering, and parameter sharing) while outperforming them.
2) For CIFAR100 & ImageNet experiments, we add new results on MobileNetV2 and ResNet50 to show that shared orthogonal bases alone without filter factorization still saves much parameters while maintaining, or further increasing, the performance (Please see Figures 5~7 in Appendix for detailed block structure.) For example, on ImageNet dataset, our ResNet50 saves 28.6% parameters while losing only 0.1% top-1 (and 0.01% top5) accuracy. Due to limited time and resources, we obtained this result without any hyperparameter tuning, and, hence, better performance could be achieved with further tuning.
3) We also improved the visibility of Table 1~3.
4) For ImageNet results (Section 4.2), we provide wall-clock speed measured on Nvidia Jetson TX2. The results are mixed. For ResNet34-derived models (ResNet34-S48U1*), although our model has lower FLOPs than the counterpart ResNet34, it takes 14% longer latency. This overhead mostly comes from performing convolution operations separately for shared and non-shared basis components. This requires further optimization to achieve better speedup. However, for ResNet50- and MobileNetV2-derived models, our parameter-efficient models show slightly lower latency (~3.3%). This is because our parameter-efficient models better make use of the limited cache and memory of the embedded device.

---

### Decision · Program_Chairs · 2021-01-07
**Final Decision**

**Decision:**

Reject

**Comment:**

I agree with the concerns raised by the reviewers. In particular, the issues of novelty and experimental evaluation (mentioned in the revision summary) remain the major weak points of the paper. My impression is that the changes made in the revision represent a significant experimental addition to the paper, one which might merit a full pass through peer review, and one which in any event did not alter the reviewers' scores. While I think this paper has something to contribute (and the empirical results suggest the method may outperform competitors), I think it would be improved by a rewrite (and possibly a restructure) that makes the part that is your contribution much more clear. For example, in the abstract, it's only in the sentence "We show both theoretically
and empirically that potential vanishing/exploding gradients problems can be mitigated by enforcing orthogonality to the shared filter bases" that we actually get to the part that is really novel about this contribution (the "enforcing orthogonality"): that would ideally be much earlier in the abstract.